# Chitosan Grafted with Thermoresponsive Poly(di(ethylene glycol) Methyl Ether Methacrylate) for Cell Culture Applications

**DOI:** 10.3390/polym15061515

**Published:** 2023-03-18

**Authors:** Natun Dasgupta, Duo Sun, Maud Gorbet, Mario Gauthier

**Affiliations:** 1Department of Chemistry, University of Waterloo, 200 University Avenue West, Waterloo, ON N2L 3G1, Canada; 2Department of Systems Design Engineering, University of Waterloo, 200 University Avenue West, Waterloo, ON N2L 3G1, Canadamgorbet@uwaterloo.ca (M.G.)

**Keywords:** chitosan, thermoresponsive, grafting, LCST, PMEO_2_MA, cell culture, RAFT, biocompatibility

## Abstract

Chitosan is a polysaccharide extracted from animal sources such as crab and shrimp shells. In this work, chitosan films were modified by grafting them with a thermoresponsive polymer, poly(di(ethylene glycol) methyl ether methacrylate) (PMEO_2_MA). The films were modified to introduce functional groups useful as reversible addition–fragmentation chain transfer (RAFT) agents. PMEO_2_MA chains were then grown from the films via RAFT polymerization, making the chitosan films thermoresponsive. The degree of substitution of the chitosan-based RAFT agent and the amount of monomer added in the grafting reaction were varied to control the length of the grafted PMEO_2_MA chain segments. The chains were cleaved from the film substrates for characterization using ^1^H NMR and a gel permeation chromatography analysis. Temperature-dependent contact angle measurements were used to demonstrate that the hydrophilic–hydrophobic nature of the film surface varied with temperature. Due to the enhanced hydrophobic character of PMEO_2_MA above its lower critical solution temperature (LCST), the ability of PMEO_2_MA-grafted chitosan films to serve as a substrate for cell growth at 37 °C (incubation temperature) was tested. Interactions with cells (fibroblasts, macrophages, and corneal epithelial cells) were assessed. The modified chitosan films supported cell viability and proliferation. As the temperature is lowered to 4 °C (refrigeration temperature, below the LCST), the grafted chitosan films become less hydrophobic, and cell adhesion should decrease, facilitating their removal from the surface. Our results indicated that the cells were detached from the films following a short incubation period at 4 °C, were viable, and retained their ability to proliferate.

## 1. Introduction

The applications of biopolymers in different areas are increasing steadily. Chitin, a polymer of acetylglucosamine units linked via 1,4-glycosidic bonds, is a structural polysaccharide found mainly in crustaceans, fungi, and yeast. Chitosan is derived from chitin via N-deacetylation to form glucosamine units, although the complete deacetylation of chitin is difficult [1,2]. The main functional groups present in chitin are primary alcohol, secondary alcohol, and primary amine groups, with some acetamide groups remaining due to incomplete deacetylation [2]. While the properties of chitosan differ depending on its molecular weight and degree of deacetylation, its exceptional film-forming ability makes it ideal for a wide range of applications, including wound healing and drug delivery [1,2]. Chitosan and chitin have been studied and used for implants and wound care for a long time, as ancient Japanese fishermen and the United States Army used powdered crab shells to treat injuries [3]. Chitosan was found to associate with a large number of mitotic cells in the wound bed, to stimulate faster epithelium growth at wound sites, and to promote collagen deposition [3]. Through increased cell attachment, chitosan has been found to contribute to cytokine and growth factor production [4,5]. Because of its polycationic nature, chitosan also has antibacterial and antifungal properties [6]. All of these factors make chitosan films excellent cell growth substrates.

Modified chitosan has likewise garnered significant attention for potential biomedical applications. One example of this is the grafting of thermoresponsive polymers, becoming hydrophobic above their lower critical solution temperature (LCST), as demonstrated by Chen et al. [7]. Solutions of chitosan grafted with poly(N-isopropylacrylamide) (PNIPAM) were indeed shown to form hydrogels after equilibrating to body temperature, making them useful as injectable hydrogels. The thermoresponsive gels were capable of entrapping chondrocytes and meniscus cells, as well as stimulating initial cell–cell interactions. Another example was the grafting of PNIPAM on thiolated chitosan by Radhakumary et al. for wound healing applications [8]. The thermoresponsive nature of PNIPAM promoted the controlled release of ciprofloxacin, protecting wounds over extended time periods. The incorporation of hydroxybutyl groups in chitosan also provided thermoresponsive materials that were investigated for cell culture and detachment [9].

The current project involved synthesizing thermoresponsive chitosan films by grafting them with a thermoresponsive polymer, so that they could serve as substrates for cell growth at 37 °C (incubation temperature). Considering that PNIPAM can potentially display cytotoxicity effects [10], we wanted to explore chitosan grafted with poly(di(ethylene glycol) methyl ether methacrylate) (PMEO_2_MA), a thermoresponsive polymer having a lower critical solution temperature (LCST) of around 26 °C [11], as a potentially biocompatible and non-cytotoxic substrate for cell culture. The chitosan films were modified to form a chitosan-based RAFT agent, followed by RAFT grafting of the di(ethylene glycol) methyl ether methacrylate (MEO_2_MA) monomer. The characteristics of the grafted PMEO_2_MA chains could be controlled by varying either the degree of substitution of the RAFT agent or the amount of MEO_2_MA added in the reaction. The aim was to facilitate cell interactions with the PMEO_2_MA-grafted chitosan films through hydrophobic interactions at 37 °C. Since PMEO_2_MA becomes less hydrophobic as the temperature drops below its LCST, the attached cells should become less adherent upon cooling, potentially enabling the PMEO_2_MA-grafted chitosan films to be reusable substrates for cell growth and detachment.

## 2. Materials and Methods

### 2.1. Chemicals and Materials

High molecular weight deacetylated chitosan (poly(D-glucosamine), molecular weight 310–375 kg/mol, >75% deacetylated), methanol (ACS reagent, ≥99.8%), N,N-dimethylformamide (DMF, HPLC, ≥99.9%), LiCl (≥99%), deuterated dimethyl sulfoxide (DMSO-*d*_6_, 99.9% atom), CS_2_ (ACS reagent, ≥99.9%), potassium persulfate (ACS reagent, ≥99.0%), sodium ascorbate (ACS reagent, ≥98%, crystals), NaOH (ACS reagent, ≥98%, pellets), methyl 2-bromopropionate (98%), di(ethylene glycol) methyl ether methacrylate (MEO_2_MA, 95%), hydrochloric acid (30% HCl *w*/*w* in H_2_O), ethanol (HPLC, 95%), aluminum oxide, and inhibitor remover columns were all purchased from Sigma-Aldrich (Oakville, ON, Canada). Dulbecco’s modified Eagle’s medium (DMEM), TrypLE Express, fetal bovine serum (FBS), penicillin–streptomycin (Pen-Strep), and Invitrogen™ LIVE/DEAD™ Viability/Cytotoxicity Kits were from Fisher Scientific (Ottawa, ON, Canada). A keratinocyte serum-free medium and supplement were from ScienCell (Carlsbad, CA, USA). The TACS XTT cell proliferation assay was from R&D Systems (Minneapolis, MN, USA). The Spectra/Por dialysis tubing with a 100–500 Da molecular weight cut-off (MWCO) range was from Spectrum Laboratories Inc. (Shewsbury, MA, USA). The inhibitors in MEO_2_MA and 2-HEA were removed with neutral alumina and inhibitor remover columns.

### 2.2. Preparation of Chitosan Films

Deacetylated chitosan (2 g, 10.5 mmol glucosamine and acetylglucosamine units) was added to deionized water (100 mL) in a 250 mL beaker, followed by acetic acid (0.6 mL, 0.01 mol). A propeller-type mechanical stirrer at 400 rpm was used to disperse the chitosan for 2 h until a gel-like consistency was achieved. To remove air bubbles from the gel, it was centrifuged at 10,000 rpm for 10 min. The gel was then evenly distributed in 40 mm × 5 mm polytetrafluoroethylene molds, each holding 15 mL of liquid. The water was allowed to evaporate for 2 days and the films were washed with a mixture of methanol and 0.1 M NaOH (1:1 by volume). The films were then dried in a vacuum oven at 60 °C. For cell studies, the films were washed with only 0.1 M NaOH and processed for sterilization in the cell culture facility (see Section 2.8.1).

### 2.3. Synthesis of Xanthated Chitosan Films

Two deacetylated chitosan films (250 mg, 0.35–0.40 mm thickness, 1.32 mmol glucosamine and acetylglucosamine units) were dipped in water (5 mL) in a 10-mL round-bottom flask (RBF) to be vortexed and stirred with a magnetic stirring bar to allow the films to swell. NaOH (80 mg, 2 mmol) was then added and the stirring was continued for 30–60 min before CS_2_ (0.20 mL, 3.31 mmol) was added dropwise. The flask was transferred to a water bath at 35 °C for 2 h, during which time the solution turned reddish-orange. The mixture was left to cool to room temperature, then the xanthated chitosan (Chito-CS_2_) films were removed from the solution, washed with a mixture of methanol and 0.1 M NaOH (1:1) several times, allowed to dry in a fume hood for 4 h, and finally left in a vacuum oven at 60 °C overnight. The films were transparent but had a light red tint. A high-DS sample was also made by doubling the amount of CS_2_ used, with all other reaction parameters held constant. For the cell studies, the films were washed with only 0.1 M NaOH and then processed for sterilization in the cell culture facility (see Section 2.8.1).

### 2.4. Synthesis of RAFT Agent

Two Chito-CS_2_ films (250 mg) were added to DI water (5 mL) in a 10-mL RBF and the mixture was stirred with a magnetic stir bar for 30 min to allow the films to swell. Methyl 2-bromopropionate (0.20 mL, 1.8 mmol) was then added dropwise and the flask was placed in a water bath at 80 °C for 40 min. The solution, which turned light yellow by the end of the reaction, was then left to cool to room temperature. The resulting chitosan-based RAFT agent (Chito-RAFT) films were removed from the solution, washed several times with a mixture of methanol and water (1:1 by volume), allowed to dry in a fume hood for 4 h, and then dried in a vacuum oven overnight at 60 °C. The Chito-RAFT films had a light-yellow color. For cell studies, the films were washed with only 0.1 M NaOH (see Section 2.8.1).

### 2.5. Synthesis of Chito-g-PMEO_2_MA

A Chito-RAFT film (110 mg, 0.35–0.40 mm thickness) was added to DI water (5 mL) in a 10 mL RBF. The mixture was stirred with a magnetic stirring bar for 30 min to allow the Chito-RAFT film to swell. Potassium persulfate (11 mg, 0.04 mmol) and MEO_2_MA (47.3 mg, 0.26 mmol, for a target PMEO_2_MA content of 15 wt %) were added to the solution, before degassing them with a steady flow of N_2_ gas for 30 min. Sodium ascorbate (9 mg, 0.045 mmol) was then added to the reaction mixture, which was placed in a water bath at 30 °C for 120 min. After the reaction, the solution was allowed to cool to room temperature, the PMEO_2_MA-grafted chitosan (Chito-*g*-PMEO_2_MA) films were removed from the solution, washed several times with a mixture of methanol and water (1:1 by volume), allowed to dry in a fume hood for 4 h, and then left in a vacuum oven overnight at 60 °C. The yield was 120 mg, with a PMEO_2_MA content of 8.7 ± 1 wt % based on the mass increase. PMEO_2_MA contents of 30 and 60 wt % were also targeted by varying the amount of MEO_2_MA added in the reaction. For cell studies, the films were washed with only 0.1 M NaOH and stored in DI water at 22 °C until reaching sterilization in the cell culture facility (see Section 2.8.1).

### 2.6. Cleavage of PMEO_2_MA Chains from Chito-g-PMEO_2_MA

A 150 mg sample of the Chito-*g*-PMEO_2_MA film was stirred with 15 mL of 1 M HCl in a 25 mL RBF. A condenser was attached and the reaction was stirred at 95 °C for 12 h. A white precipitate formed by the end of the reaction, which redissolved upon cooling in an ice-water bath. The solution of degraded chitosan (free D-glucosamine) and cleaved PMEO_2_MA chains was dialyzed for 72 h in a 100–500 Da dialysis bag. The purified solution was air-dried in a fume hood, and then in a vacuum oven at 60 °C overnight. The dry residue was dissolved in DMSO-*d*_6_ for the ^1^H NMR analysis and in DMF for the GPC characterization.

### 2.7. Material Characterization

The chitosan films were characterized at different steps of the reaction using ATR–FTIR spectroscopy, water uptake tests, and contact angle measurements. ^1^H NMR and GPC analyses were used to characterize the grafted polymer chains.

#### 2.7.1. ATR–FTIR

The chemical compositions of the films and potential interactions between their components were investigated using attenuated total reflectance–Fourier transform infrared (ATR–FTIR) spectroscopy. A PerkinElmer Spectrum Two FTIR Spectrometer was used, equipped with a Universal ATR Two accessory made of a diamond/ZnSe mixture operating with single bounce reflection. The software used was PerkinElmer Spectrum IR version 10.6.2. The spectra were generated by averaging 4 scans recorded with a LiTaO_3_ detector at a resolution of 4 cm^−1^ in the range of 4000 to 370 cm^−1^.

#### 2.7.2. ^1^H NMR

A sample of cleaved PMEO_2_MA chains (20 mg) was dissolved in 1 mL of DMSO-*d*_6_ for the analysis. The spectra were recorded at 25 °C on a 300 MHz Bruker instrument by averaging 128 scans. The reported chemical shifts were relative to the solvent protons at 2.50 ppm for DMSO-*d*_6_.

#### 2.7.3. Gel Permeation Chromatography (GPC)

The apparent (polystyrene-equivalent) number-average molecular weight (M_n_) and polydispersity index (PDI = M_w_/M_n_) values of the grafted polymer chains were determined on a Hewlett Packard 1100 HPLC system equipped with a refractive index (RI) detector and two Jordi Resolve polydivinylbenzene columns (7.8 mm internal diameter × 300 mm length). The instrument used DMF with 0.1% LiCl at a flow rate of 0.9 mL/min and a temperature of 40 °C as the mobile phase, and the samples were injected at a concentration of 3 mg/mL after filtration through 0.2 μm polytetrafluoroethylene membrane filters. Polystyrene standards were used to calibrate the instrument.

#### 2.7.4. Contact Angle Measurements

For the measurements, each film was rehydrated in 15 mL of water for 15 min. Excess water was blotted away using filter paper, and the film was flattened by placing a heavy metal block on it for 30 min. The wetting behavior of the chitosan samples was evaluated at different points of the reaction through contact angle (*θ_c_*) measurements on a customized apparatus with a built-in camera (Basler scA1000−30fm). Images were recorded every second after the deposition of an 8 µL droplet with a syringe. The photos were recorded with a custom-made LabVIEW version 14.0 program and processed with ImageJ version 1.52 Java 1.8.0_112 (64-bit) [12]. The wetting contact angles were determined using the ‘contact angle’ tool as the angles formed by the liquid at the three-phase (liquid, gas, solid) boundary, as shown in the Appendix A. This plug-in function [13] calculates the contact angle of a drop on a flat surface using a sphere approximation and an ellipse approximation. The data reported were the averages of three independent measurements, and the error of the measurements was taken as the standard deviation.

#### 2.7.5. Water Uptake Tests

Dry chitosan films (25–40 mg) with different PMEO_2_MA contents were submerged in 10 mL of deionized water at either 22 or 37 °C until equilibrium water uptake was achieved (~2 h). After removing excess water via blotting with filter paper, the mass of the wet samples was calculated. The water uptake index (WUI) was calculated using the following equation:(1)$$WUI=\left[\frac{W_{w}-W_{d}}{W_{d}}\right]*100\%$$
where W_w_ and W_d_ are the masses of the wet and dry films, respectively.

### 2.8. Cell Study with Chitosan Films

Three different cell lines were grown on modified and unmodified chitosan films, and the cell proliferation and viability were assessed for up to 7 days in culture. To further assess the cell viability and proliferation, after temperature-dependent cell detachment, the cells were seeded in tissue-culture-treated polystyrene (TCPS) wells.

#### 2.8.1. Film Sterilization

The modified and unmodified chitosan films used for the cell study were prepared by placing them in a 50 mL tube and washing them for 5 min with sterile water. After decanting the water, the samples were sterilized via washing under agitation (300 rpm) for 15 min in 70% ethanol, and again with 70% ethanol for 10 min. The samples were then washed in phosphate-buffered saline (PBS, Lonza, Cambridge, MA, USA) for 15 min and stored in PBS at room temperature in the dark until the experiments.

On the day prior to the cell experiments, to provide a consistent surface area for cell growth, discs measuring 8 mm in diameter were cut with a sterile hole hunch under aseptic conditions. The discs were placed in a 48-well TCPS plate. To promote cell adhesion, the discs of modified and unmodified chitosan films were incubated overnight at 37 °C (5% CO_2_) in 500 μL DMEM supplemented with 50% fetal bovine serum (FBS) and 1% penicillin/streptomycin (P/S) [14]. Unmodified chitosan films were used as controls.

#### 2.8.2. Cell Culture and Seeding on Films

To assess the cytotoxicity for a broad spectrum of cells, three types of immortalized cells were tested: the NIH 3T3 murine fibroblast cell line (ATCC), the RAW 264.7 mouse monocyte/macrophage cell line, and the HPV-modified human corneal epithelial cell line (HCEC, Gorbet lab). The cells were grown in a flask at 37 °C (5% CO_2_) until they were 90% confluent, and the medium was changed every 2–3 days. DMEM supplemented with 10% FBS and 1% P/S (DMEM/FBS) was used for the NIH 3T3 fibroblasts and RAW 264.7 macrophages, while the HCECs were grown in serum-free keratinocyte medium (KM) supplemented with growth factors and 1% P/S [3,15,16]. On the day of the experiment, the medium was removed and adherent cells were detached after 12 min of incubation in TrypLE Express at 37 °C. The cells were then centrifuged at 1300 rpm for 5 min. The supernatant was aspirated and the cell pellet was resuspended in 2 mL of DMEM/FBS or KM and a cell count was performed using the hematocytometer.

After removing the DMEM/50% FBS from all wells, 30,000 cells were seeded per well in a final total medium volume of 400 μL. As controls, cells were also seeded in TCPS wells with no disc. The cells were then incubated at 37 °C (5% CO_2_) for up to 7 days, with the medium changed every 2–3 days.

#### 2.8.3. Proliferation as Measured Using the XTT Assay

The cell proliferation after 1, 3, and 7 days of culture was characterized with an XTT metabolism test using sodium 3,3’-[1-(phenylaminocarbonyl)-3,4-tetrazolium]-bis(4-methoxy-6-nitro)benzenesulfonic acid hydrate. On the day of the assay, to ensure that only cell proliferation on the discs was measured, the discs were transferred to a new 48-well plate. Next, 600 μL of XTT solution was added to each well, including blank wells for the background control, and the samples were incubated for 2.5 h at 37 °C (5% CO_2_). After 2.5 h, the supernatant was transferred to a 96-well plate and the absorbance was measured at 490 nm with a reference wavelength of 630 nm on a SpectraMax Plus 384 Microplate Reader (Molecular Devices, San Jose, CA, USA). All results are expressed as relative absorbance values (corrected for the absorbance of the background control solution) [14].

#### 2.8.4. Live and Dead Assay

Following the XTT test, live/dead staining was performed. The modified and unmodified chitosan films as well as the TCPS controls were incubated with 500 μL of DMEM/FBS containing 0.25 μL of 1 mmol/L of Calcein AM (stains live cells green) and l μL of ethD-1 (stains dead cells red) for 25 min at 37 °C. The samples were then imaged with a Nikon Eclipse TS100 fluorescence microscope (Tokyo, Japan) [14].

#### 2.8.5. Temperature-Induced Cell Detachment and Viability

After imaging the NIH 3T3 fibroblasts on day 3 and the RAW 264.7 macrophages on days 3 and 7 on the films, the samples were gently washed with PBS and transferred to a new 48-well plate. DMEM/FBS was added to each well and the samples were incubated in a refrigerator (2–8 °C) for 15 min. The samples were then washed thoroughly with incubation medium and the detached cells were collected. The cell suspension was centrifuged for 5 min at 1300 rpm, the supernatant was aspirated, and the cell pellet was resuspended in 500 μL of DMEM/FBS. The cells that detached from the chitosan film were then seeded on a 24- or 48-well plate to be cultured for 3 days at 37 °C, after which time the live/dead assay was repeated.

#### 2.8.6. Statistical Analysis

All results are reported as averages ± standard deviations (SDs). To evaluate the significance of the differences in cell viability, an analysis of variance (ANOVA) was performed, followed by a post hoc Tukey test using the statistical analysis software Statistica (Tulsa, OK, USA). A *p* value of less than 0.05 was required for statistical significance. The number of experiments was at least equal to three, with experiments being run on different days.

## 3. Results and Discussion

### 3.1. Synthesis of Chitosan-Based RAFT Agent and Chito-g-PMEO_2_MA

The chitosan was modified to add thiocarbonylthio groups, serving as RAFT-active sites, as a starting point for the synthesis. This was accomplished by first reacting the hydroxyl groups on the glucosamine units with carbon disulfide in a heterogeneous reaction under alkaline conditions, namely in the presence of NaOH. The RAFT agent was obtained by reacting the xanthated chitosan intermediate with methyl 2-bromopropionate (Figure 1).

The chitosan-based RAFT (Chito-RAFT) agent obtained was then used to graft PMEO_2_MA on the film. The RAFT polymerization mechanism involves a chain transfer agent in form of a thiocarbonylthio functional group (RAFT agent) to control the molecular weight of the grafted polymer in a free radical reaction [17]. The free radicals necessary to initiate the reaction were generated with a redox system consisting of potassium persulfate and sodium ascorbate. Liu et al. used this redox initiator to graft poly(2-methoxyethyl acrylate) onto a macromolecular RAFT agent derived from poly(poly(ethylene glycol) methyl ether methacrylate). That initiator system was found to be highly efficient at 30 and 40 °C, without an induction period in either case, and 90% conversion was accomplished within 1 h with a PDI of 1.10 [18].

Attenuated total reflectance–Fourier transform infrared spectroscopy (ATR–FTIR) was used to characterize the chitosan substrate and the modified chitosan films. The carbonyl amide (C=ONHR stretch) and amine (HN-H bend) bands at 1670 and 1590 cm^−1^, respectively, are characteristic of chitosan. Since the chitosan used was >75% deacetylated, the C=ONHR stretch was used to normalize the different spectra shown in Figure 1. As expected, all four modified chitosan films had a characteristic ester (OC=O carbonyl stretch) band at 1728 cm^−1^, but the chitosan did not. Even though the results are only semi-quantitative, the OC=O band first appeared when the chitosan was modified to form the Chito-RAFT agent, and the absorbance increased for a higher targeted DS, due to the nucleophilic substitution reaction between xanthated chitosan and methyl 2-bromopropionate with bromine as the leaving group. As seen in Figure 1, due to the presence of ester groups in MEO_2_MA, the intensity of the ester stretch band increased further as the targeted grafted PMEO_2_MA content was increased from 15 to 60 wt % for the chitosan-*g*-PMEO_2_MA (Chito-*g*-PMEO_2_MA).

Since chitosan is not dispersible in most organic and aqueous solvents, the composition of the Chito-*g*-PMEO_2_MA samples was rather determined by comparing the weights of the films before (Chito-RAFT agent) and after PMEO_2_MA grafting to calculate the percentage of grafted PMEO_2_MA by weight in the product. A larger number of chains can be assumed to be grafted on or near the surfaces of the films, because the aqueous reactions were heterogeneous. However, since the films were swollen in the reaction medium before the reactions, the occurrence of grafting inside of the films could not be excluded.

The grafting efficiency was estimated using Equation (2):(2)$$Grafting\;efficiency=\left[\frac{W_{r}-W_{i}}{W_{m}}\right]*100\%$$

The PMEO_2_MA content in the product, expressed as a weight % increase relative to the initial Chito-RAFT film, was calculated from Equation (3):(3)$${\mathrm{PMEO}}_{2}\mathrm{MA}\;\mathrm{content}=\left[\frac{W_{r}-W_{i}}{W_{i}}\right]*100\%$$
where W_r_ is the weight of the recovered PMEO_2_MA-grafted chitosan film, W_i_ is the weight of the chitosan-based RAFT agent film, and W_m_ is the weight of monomer added in the grafting reaction. The average grafting efficiency and the average PMEO_2_MA content were calculated by taking the average of 3 trials for each sample type. As shown in Appendix A, the grafting reaction was not very efficient. The highest monomer grafting efficiency (49 ± 2%) was observed for Chito-*g*-PMEO_2_MA (15 wt %), which decreased to 47 ± 2% for the 30 wt % target, and further to 29 ± 3% as the target PMEO_2_MA content was increased to 60 wt %. The reaction was allowed to proceed for 2 h, during which time the Chito-*g*-PMEO_2_MA (15 wt %) and Chito-*g*-PMEO_2_MA (30 wt %) achieved 50% grafting efficiency within the experimental error limits, while the remaining chains were not grafted onto the chitosan and remained in the solution. The non-grafted chains in the solution for the Chito-*g*-PMEO_2_MA sample (60 wt %) were analyzed via GPC. The apparent M_n_ of the soluble material was determined to be 135 kg/mol, with a PDI of 1.62. As discussed in more detail in Section 3.2, these values were comparable to the apparent M_n_ and PDI values measured for the cleaved PMEO_2_MA chains of Chito-*g*-PMEO_2_MA (60 wt %). The average PMEO_2_MA contents determined from the weight measurements were 8.7 ± 1, 20 ± 1, and 43 ± 3 wt % for the 15, 30, and 60 wt % targeted PMEO_2_MA contents, respectively. As the monomer content in the grafting reaction was increased, the amount of PMEO_2_MA grafted onto the chitosan nevertheless still increased, as indicated by the experimental Chito-*g*-PMEO_2_MA compositions reported in Appendix A. The reactions were reproducible, as standard deviations of 1–3% were obtained for all three samples for the average grafting efficiency and PMEO_2_MA content of the Chito-*g*-PMEO_2_MA samples. For comparison, Tang et al. employed RAFT polymerization to graft thermoresponsive PNIPAM onto chitosan, using a RAFT agent obtained by modifying chitosan with phthalic anhydride and S-1-dodecyl-S’-(α,α’-dimethyl-α”-acetic acid) trithiocarbonate [19]. It was reported that the RAFT polymerization yielded both grafted and soluble untethered chains. A grafting efficiency of 50% was likewise reported after 10 h, suggesting that some chains remained in the solution at the end of the reaction [19].

### 3.2. Cleavage of PMEO_2_MA Chains from Chito-g-PMEO_2_MA

Using chitosan modified with functional groups acting as RAFT agents (Chito-RAFT) for the methacrylate monomer, two different strategies were developed to control the chain length and the composition of the target Chito-*g*-PMEO_2_MA films. The first strategy was to vary the degree of substitution (DS) of the chitosan-based RAFT agent, while keeping the amount of monomer used in the grafting reaction unchanged. To determine whether that approach had an impact on the chain length, the target DS for the Chito-RAFT substrate used in the synthesis of Chito-*g*-PMEO_2_MA (60 wt %) was doubled. Due to the greater number of reactive sites present in the film, a RAFT agent with a higher DS should generate a larger number of shorter polymer chains for a set amount of monomer added in the reaction. Conversely, a RAFT agent with a lower DS should generate longer polymer chains due to the presence of fewer initiating sites. The other strategy investigated was to vary the amount of monomer added to the Chito-RAFT agent for a given DS, since the length of the PMEO_2_MA chains should increase as more monomer is added in the grafting reaction. To verify this hypothesis, the target monomer (MEO_2_MA) content was varied from 15 to 60% of the Chito-RAFT film substrate weight.

The molecular weight of the thermoresponsive PMEO_2_MA chains grafted to chitosan was determined using a GPC analysis after degrading the chitosan substrate to N-acetylated and deacetylated glucosamine units using 1 M HCl. The degradation rate of chitosan increases with the acid concentration and temperature [20], and refluxing in 1 M HCl for 12 h led to the complete degradation of the chitosan component while leaving the PMEO_2_MA chains unaffected. After the reaction, the PMEO_2_MA was separated from the degraded chitosan components by dialysis for 72 h. The integrity of the cleaved PMEO_2_MA chains was verified using an NMR analysis (Figure 2). The integral of the peak for the methyl protons (3H, labeled 1) was in a 1.45:1 ratio to the methylene protons in the glycol unit next to the ester group (2H, labeled 3) or was identical, within the error limits, with the expected ratio of 1.5:1. This implies that the diethylene glycol units in the PMEO_2_MA were not cleaved during the reaction and that only the glycoside linkages were degraded. In the PMEO_2_MA sample recovered after the degradation of the chitosan component (Figure 2), an anomeric proton signal was also absent at 5.20 ppm, confirming the absence of chitosan in the product. These results were consistent with a report by Aljbour et al., who determined an average yield of 87% for the depolymerization of chitosan into monomers and dimers of glucosamine when the reaction was conducted under reflux in 2 M HCl for 24 h [21]. It was also shown that the chitosan was deacetylated during depolymerization, such that the monomers and dimers contained mostly deacetylated glucosamine units [22].

Chito-*g*-PMEO_2_MA samples with 15, 30, and 60 wt % target PMEO_2_MA contents, as well as Chito-*g*-PMEO_2_MA with a 60 wt % target PMEO_2_MA content obtained with a higher DS Chito-RAFT substrate, were studied via GPC to demonstrate control over the molecular weight of the grafted PMEO_2_MA chains as a function of the reaction conditions used (Appendix A). The apparent (polystyrene-equivalent) M_n_ of the cleaved PMEO_2_MA chains for Chito-*g*-PMEO_2_MA samples with target PMEO_2_MA contents of 15, 30, and 60 wt % were indeed 43, 85, and 128 kg/mol, respectively, with PDI values in the 1.71–1.88 range (Appendix A). Therefore, when increasing amounts of monomer were added to a constant DS Chito-RAFT substrate, the grafted PMEO_2_MA chains grew longer than expected. Furthermore, the PMEO_2_MA chains cleaved from Chito-*g*-PMEO_2_MA (60 wt %, high-target DS) had an apparent M_n_ of 105 kg/mol and a PDI of 1.88 (Appendix A), somewhat lower than for the low DS Chito-*g*-PMEO_2_MA (60 wt %) sample (M_n_ = 128 kg/mol). Due to the increased DS, shorter PMEO_2_MA chains were grown from the chitosan substrate than expected, albeit the variation in M_n_ was not as pronounced as in the situation where the amount of monomer added was varied. Jiang et al. synthesized a chitosan-based RAFT agent to graft poly(N-isopropylacrylamide) (PNIPAM) and obtained a PDI of 1.28 for M_n_ = 16.2 kg/mol [23]. The molecular weights attained for the PMEO_2_MA chains were much higher in the current investigation, but the PDI values were also higher.

### 3.3. Contact Angle Measurements

The sessile water drop contact angle test was used to characterize the hydrophilic or hydrophobic nature of the modified chitosan films above and below the LCST of the grafted PMEO_2_MA chains. Hydrophilic surfaces are generally considered to have water contact angles below 90°, while hydrophobic surfaces have contact angles above 90° [24]. The contact angle measurement results are compared for selected Chito-*g*-PMEO_2_MA samples with different target thermoresponsive polymer contents in Figure 3, and the quantitative results are summarized in Appendix A. A film of PMEO_2_MA synthesized by radical polymerization (M_n_ = 60.4 kg/mol, PDI = 2.5) cast on a glass slide was also used as a control for comparison with the Chito-*g*-PMEO_2_MA samples. The experiment could not be carried out on unmodified chitosan films because they were too hydrophilic and readily absorbed the water droplets. Not surprisingly, PMEO_2_MA was found to be hydrophobic at 40 °C, above its reported LCST of 26 °C [25]. An increase in hydrophobicity can also clearly be seen for the different Chito-*g*-PMEO_2_MA samples as their PMEO_2_MA content increases. The presence of grafted PMEO_2_MA on the surface should reduce the charge density of chitosan, resulting in lower hydrophilicity [26]. Interestingly, the contact angles of all the Chito-*g*-PMEO_2_MA samples were significantly higher than for the PMEO_2_MA homopolymer reference, suggesting enhanced surface hydrophobicity for these films. This may be explained in part by the much greater M_n_ of the cleaved PMEO_2_MA chains in the 30 and 60 wt % samples than for the reference sample, which had a high contact angle even under the LCST. For the 30 and 60 wt % Chito-*g*-PMEO_2_MA samples, the contact angle measurements may have been lower at temperatures <22 °C, but the Chito-*g*-PMEO_2_MA samples tended to form gels when stored in the refrigerator (4 °C) overnight. Furthermore, it was impossible to undertake contact angle measurements below 22 °C (room temperature) due to the lack of a cooling system on the instrument.

In the low temperature range (22 °C, below the LCST of the PMEO_2_MA chains), the contact angles also varied in the order of Chito-*g*-PMEO_2_MA (15 wt %) < Chito-*g*-PMEO_2_MA (30 wt %) < Chito-*g*-PMEO_2_MA (60 wt %), especially at longer measurement times. This trend can be rationalized in terms of the decreasing influence of the chitosan component of the samples as their PMEO_2_MA content increased. Noticeably, the longer PMEO_2_MA chains grafted in the 30 and 60 wt % samples resulted in less drastic changes in contact angle, which may be related to the long PMEO_2_MA chains shielding the chitosan component more effectively from water than in the 15 wt % sample. Furthermore, as seen in Appendix A, contact angle changes at 40 °C were insignificant within error limits over the 4 s time scale of the measurements, while a small decrease was observed for the Chito-*g*-PMEO_2_MA samples at 22 °C. This s presumably due to the rearrangement of the hydrophilic chitosan component to contact the water phase, which is facilitated by the increased mobility of the hydrated PMEO_2_MA chains at low temperature.

Finally, there was an obvious increase in the contact angles for all Chito-*g*-PMEO_2_MA samples as the temperature was increased from 22 °C (below the LCST of PMEO_2_MA) to 40 °C (above the LCST), as expected for these thermoresponsive materials. Plunkett et al. similarly observed that the water contact angle increased from 68.4° ± 0.4° to 79° ± 1° for PNIPAM monolayers self-assembled on gold surfaces as the temperature was increased from below to above the LCST of PNIPAM [27]. The magnitudes of the contact angle changes observed for the Chito-*g*-PMEO_2_MA 15 and 30 wt % samples were larger than for the PNIPAM monolayers, presumably due to decreased shielding of the chitosan component by the grafted PMEO_2_MA chains.

### 3.4. Water Uptake Tests

The water uptake properties of modified and unmodified chitosan films are compared in Figure 4. The water uptake index (WUI) of chitosan, expressed as the % water uptake relative to the weight of the dry samples, decreased substantially with the amount of PMEO_2_MA grafted to the chitosan, particularly for target PMEO_2_MA contents of 30 and 60 wt %. However, it is necessary to be cautious in making comparisons due to variability in the measurements leading to large error bars. Detailed information regarding the WUI values and their variability are presented in Appendix A. The combination of hydrophilic hydroxyl and amino groups in the chitosan favors hydrogen bonding with water, leading to the highest WUI values being observed for chitosan and Chito-RAFT at both 22 and 37 °C. Grafting PMEO_2_MA on the chitosan films decreased their average hydrophilicity to some extent, such that the bulk water uptake of the films was decreased significantly at the higher PMEO_2_MA contents due to the decreasing contribution of the hydrophilic chitosan component. The thermoresponsiveness of the samples manifested itself as a substantial decrease in WUI at 37 °C, above the LCST of PMEO_2_MA, even though the error bars overlapped for Chito-*g*-PMEO_2_MA (60%). This was clearly due to the grafted thermoresponsive PMEO_2_MA chains accounting for a significant portion of the sample weight, becoming hydrophobic at the higher temperature and making the bulk material less hydrophilic on average. As mentioned earlier, the LCST of the chains at the surface of the Chito-*g*-PMEO_2_MA (30%) and Chito-*g*-PMEO_2_MA (60%) films could potentially be below 22 °C, which would explain why no dramatic changes in WUI were observed as the temperature was increased from 22 to 40 °C.

### 3.5. Cell Adhesion, Viability, and Proliferation on Chitosan Films

The cell culture experiments were performed with RAW 264.7 macrophages, NIH 3T3 fibroblasts, and immortalized human corneal epithelial cells (HCEC). The cells were allowed to proliferate on polystyrene plates treated for tissue culture (TCPS) and unmodified chitosan films as controls, as well as on chitosan films grafted with 30 and 60 wt % PMEO_2_MA, for up to 7 days.

The live/dead assay provides a qualitative evaluation of the cytotoxicity of a biomaterial, with the Calcein AM staining live cells green and EthD-1 staining dead cells red, which can be observed on a fluorescent microscope. Overall, the aim was to evaluate the cytotoxicity of the films towards these cells. It is important to understand that while a material might not be cytotoxic, this does not necessarily imply that it is biocompatible. Biocompatibility can be defined as the ability of a material (e.g., the chitosan films) to perform its desired function without inducing any local or systemic adverse response [28]. This pilot study aimed to characterize cell adhesion and proliferation relating to the cytocompatibility, as the cell phenotype after interactions with the material was not characterized. The thermoresponsive properties of the modified chitosan films were evaluated with RAW 264.7 macrophages and NIH 3T3 fibroblasts. This was accomplished by lowering the temperature as described in Section 2.8.5, to detach the growing cells from the films. The cells were then cultured for 3 days on a tissue culture plate, to determine their ability to adhere and be viable following detachment from chitosan. Calcein AM and EthD-1 staining methods were used again to determine the live and dead cells, respectively.

#### 3.5.1. Macrophage Interactions with Chitosan Films

The RAW 264.7 macrophages, originating from the Abelson leukemia virus derived from BALB/c mice, are capable of performing pinocytosis and phagocytosis by producing nitric oxide [29]. Macrophages, in comparison to other types of cells, are difficult to remove from any film surface; consequently, they were selected as a relevant cell type for the cell adhesion experiments [30]. As seen in Figure 5a, the RAW 264.7 macrophages proliferated well on both modified and unmodified chitosan samples, as demonstrated by the increase in absorbance values in the XTT assays between day 1 and day 7. The cell viability and proliferation were further confirmed with the live/dead assay (Figure 5c); a significant increase in cell density can be observed between day 1 and day 7. It is important to note that despite the fact that an almost confluent layer of macrophages could be seen on both the TCPS and Chito-*g*-PMEO_2_MA (30 wt %), the XTT absorbance values for Chito-*g*-PMEO_2_MA (30 wt %) were about 30% of those for TCPS (Figure 5b). This is because the metabolized dye sorbed (adsorbed or absorbed) on the films, meaning its concentration in the supernatant was reduced. Indeed, it can be seen in Appendix A that as the XTT solution was metabolized, it was sorbed on the chitosan films, meaning the values reported at day 7 underestimate the actual cell proliferation level. This was further supported by fluorescence microscopy (Figure 5c), as the cell proliferation at day 7 on Chito-*g*-PMEO_2_MA (30 wt %) appeared to be very similar to the control TCPS. The results from the live/dead assay (Figure 5c, day 1) and XTT assay also suggest that macrophages adhere well on modified chitosan films (similar level to TCPS at day 1–Figure 5b) and can proliferate well (Figure 5a). At day 1, more dead cells were observed on the Chito-*g*-PMEO_2_MA (60 wt %) (red-stained cells) as compared with the Chito-*g*-PMEO_2_MA (30 wt %), indicating the initial cytotoxicity of the Chito-*g*-PMEO_2_MA (60 wt %) (Figure 5c). However, the number of dead cells was negligible by the end of day 7. The live/dead results indicate that the RAW 264.7 macrophage cells proliferated well on the modified chitosan films, reaching a confluence level similar to the TCPS control, especially for the Chito-*g*-PMEO_2_MA (30 wt %). The lower adhesion and proliferation of the RAW 264.7 macrophages observed on the unmodified chitosan was in agreement with the prior work by Norowski et al., who demonstrated that while the unmodified chitosan was non-inflammatory (reduced nitric oxide response), the proliferation was lower as compared to TCPS [15].

Overall, the results indicate that PMEO_2_MA-grafted chitosan films are excellent surfaces for the growth of macrophages, displaying limited and no cytotoxicity for Chito-*g*-PMEO_2_MA (60 wt %) and Chito-*g*-PMEO_2_MA (30 wt %), respectively. As seen in Figure 5c, the cells appear to grow in large clumps on both chitosan and Chito-*g*-PMEO_2_MA (60 wt %), with bare areas (day 3 for Chito-*g*-PMEO_2_MA (60 wt %), day 7 for chitosan), suggesting that these films may not be as cytocompatible as Chito-*g*-PMEO_2_MA (30 wt %). Physicochemical properties such as the surface roughness are important to optimize cell adhesion. Zan et al. studied the effects of the surface roughness of chitosan microspheres on cell adhesion and found that mouse MC3T3-E1 osteoblasts preferred rough surfaces over smooth ones [31]. An increase in the PMEO_2_MA content from 30 to 60 wt % may have led to increased film roughness, reducing the cytocompatibility of the Chito-*g*-PMEO_2_MA (60 wt %) films.

#### 3.5.2. Fibroblast Interactions with Chitosan Films

Fibroblasts are among the most common cell types used in cytotoxicity studies [32]. Chitosan and *O*-carboxymethyl chitosan films were found previously to enhance the adhesion and proliferation of fibroblasts [33]. Cells can attach to the surface via specific adhesion proteins present in fetal bovine serum (FBS) or through direct interactions with chitosan [33]. It was found that while the initial adhesion of fibroblasts on modified chitosan films was about 70% relative to TCPS (Figure 6b), the fibroblasts did not proliferate well on the chitosan films, as reflected in the XTT results (Figure 6a,b), where proliferation decreased drastically from day 1 to day 7. Lower numbers of adherent NIH 3T3 fibroblasts were also observed on the unmodified chitosan films as compared with both the TCPS and modified chitosan on day 1 after seeding (Figure 6a,b, day 1). A significant reduction in XTT absorbance was observed at day 7, which was due to XTT sorbing on the chitosan, as mentioned in Section 3.5.1. The cells appeared to proliferate to some extent on the films and to stay alive over the 7 day culture period (Figure 6c). On the modified chitosan films, the adherent NIH 3T3 fibroblasts did not consistently display the spindle cell morphology typical for fibroblasts, which was observed on both the unmodified chitosan and TCPS substrates. While the modified chitosan films were not cytotoxic (no red-stained dead cells observed), clearly they were not as cytocompatible with NIH 3T3 fibroblasts as the RAW 264.7 macrophages, as they did not support cell proliferation. The cell adhesion, proliferation, and morphology have been shown to depend on the presence of proteins and the properties of the surfaces, namely the topography, roughness, and stiffness of the material [31,32]. It is currently unclear which of these factors played a role in the poor fibroblast cell interactions observed on both the modified and unmodified chitosan films. Differences in cytocompatibility depending on the cell lineage have also been reported in other studies [3,34]. Further work will be required to identify the parameters that govern these interactions to support the development of this thermoresponsive substrate.

#### 3.5.3. Human Corneal Epithelial Cell Interactions with Chitosan Films

The HPV-modified human corneal epithelial cell line (HCEC), grown in serum-free medium, was selected to assess the cytotoxicity of the chitosan films because it is highly sensitive to external stresses, such as the presence of undesirable compounds or byproducts on the film surface [35]. At day 1 after seeding, the modified films had a lower number of adhering HCECs than the unmodified chitosan films (Figure 7, day 1). After 3 days, it appeared that the HCECs were not able to proliferate as much on the 30 and 60 wt % Chito-*g*-PMEO_2_MA substrates, and a few dead (red-stained) cells could also be noticed on the Chito-*g*-PMEO_2_MA (60 wt %). The unmodified chitosan films, on the other hand, displayed enhanced cell proliferation over days 1–3. Benhabbour et al. studied the effects of hydrophilic hydroxyl-terminated dendrons and PEG on HCEC adhesion to modified gold surfaces [36]. The authors reported that the presence of oxyethylene moieties in the PEG resulted in cell-resistant surfaces, due to the absence of proteins promoting cell adhesion, but when the PEG chains were end-modified with the hydrophilic dendrons, the high density of the peripheral hydroxyl groups promoted HCEC adhesion and proliferation [36]. The situation may be comparable for the modified chitosan films, in that the high density of grafted PMEO_2_MA chains (containing oxyethylene units) on the surfaces may have reduced the exposure of hydroxyl and amino groups from the chitosan, reducing the ability of the HCECs to adhere and proliferate.

It should be noted that this observation is specific to HCEC, and highlights how the cytocompatibility of films can differ between cell lines such as macrophages RAW 264.7 and fibroblasts NIH 3T3, as discussed in the previous sections.

### 3.6. Temperature-Induced Cell Detachment and Viability

Chen et al. synthesized chitosan films grafted with poly(acrylic acid) and PNIPAM to make them thermoresponsive, and examined the cell adhesion and detachability with fibroblast cells. The films were found to be non-cytotoxic and allowed cell detachment as the temperature was lowered [37]. Additionally, Lu et al. recently reported the synthesis of thermoresponsive PNIPAM-grafted chitosan/heparin composites displaying good cytocompatibility with the murine embryonic fibroblast C3H10T1/2 cell line; the cell viability and proliferation results after detachment are pending [38]. In this investigation, the modified chitosan films were incubated for 15 min at 2–8 °C to test their thermoresponsive properties, with an unmodified chitosan film used as the control, to determine whether the cells would detach from the surfaces upon cooling, be viable, and continue to proliferate. Our own experiments were performed with RAW 264.7 (cells detached on day 3) and NIH 3T3 (cells detached on day 3 and day 7) cells. The cells that detached from the films were then cultured on TCPS for up to 3 days, after which live/dead staining was performed. As seen in Figure 8, the cells that detached from the films were viable, with no observable cell death. Unfortunately, it was not possible to perform quantitative comparisons using the XTT assay before and after cell detachment, as the chitosan films sorbed the dye, making it impossible to draw reliable conclusions (Appendix A). While the current analysis is only qualitative, it appears that Chito-*g*-PMEO_2_MA (60 wt %) may yield a slightly larger number of cells after day 3 of proliferation than Chito-*g*-PMEO_2_MA (30 wt %) (Figure 8). As discussed earlier, several factors may play a role in the interactions of chitosan with the cells, among which the amino groups in the chitosan may be important. According to Freier et al., the adhesion of dorsal root ganglion cells on chitosan was reduced as the acetylation level increased [39]. Chitosan with only 0.5% acetyl groups had the highest cell viability, approximately 8-fold higher than chitosan with 11% residual acetyl groups [39]. In the current case, the presence of the PMEO_2_MA chains likely also affected the cell adhesion and detachment. Since the Chito-*g*-PMEO_2_MA (60 wt %) contained longer polymer chains than the Chito-*g*-PMEO_2_MA (30 wt %), its hydrophobic interactions with the cells at 37 °C (above its LCST) may have increased the impact of the change in hydrophobicity relative to the shorter PMEO_2_MA chains in the Chito-*g*-PMEO_2_MA (30 wt %), and improved the ease of detachment during refrigeration. However, it should be noted that hydrophobic interactions are only one of the factors contributing to cell adhesion, in addition to other parameters such as fibronectin in FBS [40] and the film roughness [31].

## 4. Conclusions

The synthesis of PMEO_2_MA-grafted chitosan by a RAFT mechanism was successfully demonstrated. The length of the PMEO_2_MA segments can be controlled by varying the DS of the Chito-RAFT substrate or the monomer content in the reaction. It can be assumed that PMEO_2_MA grafting took place throughout the films rather than exclusively on their surface, but the presence of a concentration gradient, with the formation of a PMEO_2_MA-rich layer at or near the surface, would be expected on the basis of RAFT site accessibility in the grafting reaction. Despite the successful grafting of PMEO_2_MA on chitosan and the promising preliminary results obtained for cell adhesion, culture, and detachment, the characteristics of these films still need to be optimized to improve their performance for cell culture applications.

The PMEO_2_MA-grafted chitosan films exhibited low toxicity towards RAW 264.7 macrophages, NIH 3T3 fibroblasts, and HCECs, with Chito-*g*-PMEO_2_MA (30 wt %) displaying the highest cytocompatibility level for RAW 264.7 cells, while Chito-*g*-PMEO_2_MA (60 wt %) provided the easiest cell detachment. To better understand the impact of the PMEO_2_MA chains on the interactions of the films with different cell lines, more robust methods are required to quantify the cell growth and detachment. Moreover, a comprehensive investigation of the cell functionality and phenotype after detachment and proliferation is necessary to quantify the biocompatibility of the Chito-*g*-PMEO_2_MA substrates more accurately, to determine the potential of these films for regenerative medicine and tissue engineering applications.

## Data Availability

Available upon request.

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
