# Peer review of "Chitosan Grafted with Thermoresponsive Poly(di(ethylene glycol) Methyl Ether Methacrylate) for Cell Culture Applications"

_polymers, 2023, doi:10.3390/polym15061515_

Round 1
Reviewer 1 Report
Dear Authors
The presented work involved synthesizing thermoresponsive chitosan films, by grafting with a thermoresponsive polymer, so that they could be used as substrates for cell growth at human body temperature. using a potentially biocompatible polymer, poly(di(ethylene glycol) methyl ether methacrylate) (PMEO2MA), as the thermoresponsive component, for grafting of chitosan, benefiting from its lower critical solution temperature (LCST). The chitosan films were first modified to form a chitosan-based RAFT agent, followed by RAFT grafting of the di(ethylene glycol) methyl ether methacrylate (MEO2MA) monomer. The authors tried to control the characteristics of the grafted PMEO2MA chains by varying either the degree of substitution of the RAFT agent or the amount of MEO2MA added to the reaction. The aim was to facilitate cell interactions with the PMEO2MA-grafted chitosan films through hydrophobic interactions at 37 C. Since PMEO2MA becomes less hydrophobic as the temperature drops below its LCST, the attached cells should become less adherent upon cooling, potentially enabling the PMEO2MA-grafted chitosan films to be reusable substrates for cell growth and detachment.
The following comments and maybe suggestions may help in increasing the clarity of the work;
1. Introduction
The introduction is very short and provides no data about the use of modified chitosan with other thermosensitive polymers or other reagents in the field of study. Much information must be provided in this regard.
2.4. Synthesis of RAFT Agent
"Two Chito-CS2 films (250 mg) were added to DI water (5 mL) in a 10-mL RBF ......", what is the "RBF"?. Please clarify.
2.7.5. Swelling Tests
"Swelling" is a change in volume. The authors studied the water uptake (%). Please change the title of the section.
General comment
Why the authors did not try the homogeneous grafting technique?
In conclusion, the presented work is promising and can be recommended for publication after minor revision.
Greetings
Author Response
Dear Authors
The presented work involved synthesizing thermoresponsive chitosan films, by grafting with a thermoresponsive polymer, so that they could be used as substrates for cell growth at human body temperature. using a potentially biocompatible polymer, poly(di(ethylene glycol) methyl ether methacrylate) (PMEO2MA), as the thermoresponsive component, for grafting of chitosan, benefiting from its lower critical solution temperature (LCST). The chitosan films were first modified to form a chitosan-based RAFT agent, followed by RAFT grafting of the di(ethylene glycol) methyl ether methacrylate (MEO2MA) monomer. The authors tried to control the characteristics of the grafted PMEO2MA chains by varying either the degree of substitution of the RAFT agent or the amount of MEO2MA added to the reaction. The aim was to facilitate cell interactions with the PMEO2MA-grafted chitosan films through hydrophobic interactions at 37 °C. Since PMEO2MA becomes less hydrophobic as the temperature drops below its LCST, the attached cells should become less adherent upon cooling, potentially enabling the PMEO2MA-grafted chitosan films to be reusable substrates for cell growth and detachment.
The following comments and maybe suggestions may help in increasing the clarity of the work;
- Introduction
The introduction is very short and provides no data about the use of modified chitosan with other thermosensitive polymers or other reagents in the field of study. Much information must be provided in this regard.
More information on investigations of chitosan modified to be thermoresponsive and their applications has been added in the Introduction (page 2, lines 51-69).
2.4. Synthesis of RAFT Agent
"Two Chito-CS2 films (250 mg) were added to DI water (5 mL) in a 10-mL RBF ......", what is the "RBF"?. Please clarify.
The acronym RBF, for round-bottom flask, was defined in the paragraph just above the one to which the Reviewer is referring.
2.7.5. Swelling Tests
"Swelling" is a change in volume. The authors studied the water uptake (%). Please change the title of the section.
We thank the Reviewer for the comment. Since the measurements were done on a weight basis (rather than by measuring volume changes), the terminology used was changed to refer to the Water Uptake Index (WUI).
Why the authors did not try the homogeneous grafting technique?
We did not attempt homogeneous grafting because the goal was to graft the thermoresponsive polymer segments mainly on the surface of the chitosan films. This approach would maintain the integrity of the chitosan films and modify mainly its surface to be thermoresponsive for the cell growth application. This approach also avoids potential issues such as dissolution of the films at low temperatures, by relying upon residual crystallinity of the chitosan component to maintain film integrity.
In conclusion, the presented work is promising and can be recommended for publication after minor revision.
Greetings
Reviewer 2 Report
I have enjoyed reading the manuscript of Dasgupta et al. This is an great-designed study on the synthesis of a chitosan graft copolymer with increased hydrophobicity and thermal sensitivity. The authors propose to use the copolymer as a controlled adhesion material for growing cells. The paper shows that the approach proposed by the authors for controlling adhesion works, but in general it needs to be improved. Still, this is an excellent study showing new uses for stimulus-responsive polymers.
I have only two comments for the authors.
1. Why do the authors graft hydrophobic PMEO2MA chains onto a chitosan film rather than make films from an already obtained copolymer?
2. Introduction should be enhanced to reflect application variety of chitosan derivatives and copolymers, especially in fields of biological objects’ immobilization, wound healing, tissue engineering, etc.
Author Response
I have enjoyed reading the manuscript of Dasgupta et al. This is an great-designed study on the synthesis of a chitosan graft copolymer with increased hydrophobicity and thermal sensitivity. The authors propose to use the copolymer as a controlled adhesion material for growing cells. The paper shows that the approach proposed by the authors for controlling adhesion works, but in general it needs to be improved. Still, this is an excellent study showing new uses for stimulus-responsive polymers.
I have only two comments for the authors.
Why do the authors graft hydrophobic PMEO2MA chains onto a chitosan film rather than make films from an already obtained copolymer?
We did not attempt homogeneous grafting because the goal was to graft the thermoresponsive polymer segments mainly on the surface of the chitosan films. This approach would maintain the integrity of the chitosan films and modify mainly its surface to be thermoresponsive for the cell growth application. This approach also avoids potential issues such as dissolution of the films at low temperatures, by relying upon residual crystallinity of the chitosan component to maintain film integrity.
- Introduction should be enhanced to reflect application variety of chitosan derivatives and copolymers, especially in fields of biological objects’ immobilization, wound healing, tissue engineering, etc.
More information on investigations of chitosan modified to be thermoresponsive and their applications has been added in the Introduction (page 2, lines 51-69).
Reviewer 3 Report
1. Does the molecular weight of chitosan affect the state of the cells? Does high or low molecular weight chitosan favor cell culture in this study?
2. Section 2, clear information about the number of repetitions and statistical evaluation of the data is missing. This is crucial for high-quality research work (refering 10.1016/j.carbpol.2020.117213).
3. Figure 6C, the cell background photo of sample A on the third day is a bit reddish. The background of the photo should be consistent with the other samples.
4. The Conclusion section should be clear and concise: the important results and main conclusions drawn in this paper should be highlighted and presented in more precise language.
5. Mechanism descriptions for cell culture can be strengthened by citing 10.1021/acs.jafc.9b06120; 10.1016/j.carbpol.2020.116160 and what are the advantages of the current work compared to published articles?
6. What is the main mechanism by which the chitosan film prepared in this article promotes cell proliferation? Does unmodified chitosan also have this property?
Author Response
- Does the molecular weight of chitosan affect the state of the cells? Does high or low molecular weight chitosan favor cell culture in this study?
Research on the effects of chitosan on cells as a function of its molecular weight (MW) has been limited, as most studies focused on the degree of deacetylation (DDA). The MW and DDA are not directly correlated, and information on both parameters is not always quoted in publications, which makes comparisons difficult. Some groups have reported improved viability and proliferation for higher molecular weights and higher DDA levels, while others reported limited differences or opposite results. Such differences can be explained by the fact that the effects of chitosan on cells appear to be lineage-dependent, but also on other parameters such as the source of the chitosan used. The results of our study, which used high molecular weight chitosan, seem to be in line with prior work using fibroblasts and RAW cells, showing that unmodified chitosan supported attachment, but at lower levels than polystyrene treated for tissue culture. Considering the controversy that exists in that area, and the fact that we used the same chitosan feedstock for the whole investigation, we prefer not to discuss the potential influence of the chitosan MW on the results obtained.
- Section 2, clear information about the number of repetitions and statistical evaluation of the data is missing. This is crucial for high-quality research work (refering 10.1016/j.carbpol.2020.117213).
We appreciate the comment from the reviewer. While we had indicated repetitions in the figures, we omitted provide information on statistical analysis of the data. This has been corrected by adding Section 2.8.6 (Statistical Analysis) on page 7 of the revised manuscript. We have not added the suggested reference, as statistical analysis is common in cell studies and we used common statistical analysis tools.
- Figure 6C, the cell background photo of sample A on the third day is a bit reddish. The background of the photo should be consistent with the other samples.
Thank you for this suggestion. The picture of Figure 6C has been modified to be consistent with the other samples.
- The Conclusion section should be clear and concise: the important results and main conclusions drawn in this paper should be highlighted and presented in more precise language.
The Conclusions section has been modified to give a more concise and precise overview of the results obtained, and an outlook on future research.
- Mechanism descriptions for cell culture can be strengthened by citing 10.1021/acs.jafc.9b06120; 10.1016/j.carbpol.2020.116160 and what are the advantages of the current work compared to published articles?
We appreciate the reviewer’s suggestion to strengthen the methodologies section by providing references to prior work. The suggested references did not use the same cell lines nor viability tests, and since each cell line has a specific culture protocol, to stay in line with the reviewer’s suggestion, we have added references 3 and 14-16 using cell lines and viability tests similar to the ones in our study. We also added discussion (on page 14, line 557; page 15, line 607; page 17, line 657) with these, and additional references (34 and 38) to other work in support (on page 15, line 607; page 17, line 659) to compare our current results with prior work using chitosan and/or thermoresponsive polymers for cell detachment. The aim of our study was not towards tissue engineering, so we do not feel that comparing with the suggested articles is appropriate in the context of cell culture on chitosan and their detachment.
- What is the main mechanism by which the chitosan film prepared in this article promotes cell proliferation? Does unmodified chitosan also have this property?
At this stage of the study, the mechanisms governing cell proliferation on the substrates have not been identified. As mentioned in the manuscript (page 15, line 603), several factors can contribute to cell proliferation such as surface roughness, stiffness, proteins present at the interface, and the ability of cells to deposit their own extracellular matrix on the substrate. Additional experiments will be required to identify these mechanisms and we hypothesize that this is likely a multi-factorial effect, possibly involving fibronectin, as it has previously been shown to play an important in cell adhesion and proliferation on chitosan-based materials.
Unmodified chitosan was found to support cell proliferation, but to a different extent, and this was also found to vary with the cell type (macrophages versus fibroblasts, for example, as shown in Figures 5b and 6b). Our results are in line with prior work showing that chitosan can support the adhesion and proliferation of different cell lines, and that modifications can further improve that property, as shown in our studies with Chito-g-PMEO2MA (30 and 60 wt %).